# Longitudinal Associations of Work Stress with Changes in Quality of Life among Patients after Acute Coronary Syndrome: A Hospital-Based Study

**DOI:** 10.3390/ijerph192417018

**Published:** 2022-12-18

**Authors:** Luqiao Wang, Yunke Shi, Zhao Hu, Yanyan Li, Yan Ang, Pan Jing, Bangying Zhang, Xingyu Cao, Adrian Loerbroks, Jian Li, Min Zhang

**Affiliations:** 1Cardiology Department, The First Affiliated Hospital of Kunming Medical University, Kunming 650032, China; 2Institute of Occupational, Social and Environmental Medicine, Centre for Health and Society, Faculty of Medicine, University of Düsseldorf, 40225 Düsseldorf, Germany; 3Department of Environmental Health Sciences, Fielding School of Public Health, University of California Los Angeles, Los Angeles, CA 90095, USA; 4School of Nursing, University of California Los Angeles, Los Angeles, CA 90095, USA

**Keywords:** job strain, quality of life, effort–reward imbalance, acute coronary syndrome, longitudinal study

## Abstract

(1) Background: Targeting a sample of Chinese employees in this study, the correlation of work stress with changes in quality of life (QoL) was explored subsequent to acute coronary syndrome (ACS). (2) Methods: Patients suffering from the first ACS episode, with regular paid work before ACS, were eligible for this one-year longitudinal study. Effort–reward imbalance (ERI), together with job strain (JS) models, were employed to evaluate work stress before discharge, and QoL prior to discharge (baseline), as well as at 1, 6, and 12 months following discharge, were measured using the 8-Items Short Form (SF-8), in addition to the Seattle Angina Questionnaire (SAQ). Moreover, generalized estimating equations were used to determine the relationship of work stress to longitudinal QoL variations. (3) Results: After adjusting for covariates, high work stress at the baseline measured by JS was associated with the slow recovery of both mental health (*p* < 0.01) and physical health (*p* < 0.05) in SF-8, while ERI-measured work stress was related to slower improvement in SF-8 physical health (*p* < 0.001), SAQ-angina stability (AS) (*p* < 0.05), SF-8 mental health (*p* < 0.001), and SAQ-angina frequency (AF) (*p* < 0.05). After mutual adjustment for JS and ERI, high work stress as assessed by JS displayed no correlation with any QoL alteration (all *p* > 0.05), whereas ERI-determined work stress at a high level still presented a relationship to slow improvement in SF-8 physical health, SAQ-AS, SF-8 mental health, and SAQ-AF (all *p* < 0.05). (4) Conclusion: Work stress was associated with slow recovery of QoL in patients with ACS across one year. For ACS patients, ERI was a stronger predictor of QoL variations than JS.

## 1. Introduction

The occurrence, development and prognosis of coronary heart disease (CHD), which is considered to be a psychosomatic illness, are correlated with psychosocial factors [1]. One of these factors is work stress, meaning the “harmful physical and emotional responses that occur when the requirements of the job do not match the capabilities, resources, or needs of the worker” [2]. The adverse effects of work stress on cardiovascular diseases, especially CHD, have attracted widespread public attention over the last thirty years [3,4]. Work stress can be conceptualized according to various theoretical models. The two most dominant and most widely used models in research and practice throughout the past decades have been the job strain (JS) model and the effort–reward imbalance (ERI) model. To be specific, according to the postulation of the former model, JS is attributed to the combined actions of high job demand and low job control [5], while nonreciprocal social exchange (characterized by high cost but low gain in the workplace) is emphasized in the latter model, resulting in emotional distress status [6]. Research has repeatedly shown that, for apparently healthy workers, both JS and ERI have relationships to a 20% increased risk of the first CHD event [7,8,9]. Furthermore, for employees possessing a history of CHD, the risk of subsequent cardiac events is increased by 65% when exposed to work stress, as suggested by our previous meta-analysis [10].

The mechanisms by which work stress is linked with recurrent CHD need to be clarified. Clinically, it has been observed that, subsequent to acute coronary syndrome (ACS), impairment of quality of life (QoL) is commonly due to myocardial function alteration, inadequate adaptation to stress, and specific adaptations to one’s life style [11]. In turn, poor QoL is an independent predictor of poor prognosis of CHD, including recurrent CHD, all-cause/cardiovascular mortality in the long term, and hospital readmission [12,13,14]. Therefore, lower levels of QoL might be one plausible explanation for the potentially causal association of work stress with adverse cardiac events among workers after the first onset of ACS. Only one prior investigation, to our knowledge, which was carried out in Germany, examined the contribution of ERI-related work stress to QoL changes among employees post-acute myocardial infarction (MI) [15]. The results indicated that high baseline ERI was related to unsatisfactory physical QoL plus poor mental QoL during the follow-up process at 1 year after heart attack.

In the single prior study [15], only ERI model was used to access work stress, and the participants of the study were from a Western country (i.e., Germany). The aim of our study was to examine prospective links of work stress with changes in QoL among patients after ACS, by expanding insights from the single prior study [15] in two ways. First, in addition to the ERI we examined the JS model and compared the two models of work stress from the angle of their explanatory contributions. Those analyses can inform the development of interventions that target specific psychosocial work characteristics to improve QoL in patients ACS. Second, novel research from an Eastern country (i.e., China) is provided by our study, highlighting obvious cultural differences in work stress [16].

## 2. Materials and Methods

### 2.1. Population and Design of the Study

Patients treated in a large general hospital, the First Affiliated Hospital of Kunming Medical University, China, from March 2018 to December 2019 were recruited. Patients suffering from their first ACS episode who had regular paid work before the ACS episode were eligible. ACS is defined as incorporating multiple acute myocardial ischemic states, such as ST-segment elevation myocardial infarction (STEMI) [17], non-ST segment elevation myocardial infarction (NSTEMI), and unstable angina (UA) [18]. Percutaneous coronary intervention was performed for all patients according to the routine standard ACS treatment. Demographic and clinical data were obtained from patients’ medical records, while work stress, other relevant psychosocial factors (including anxiety, depression, and job burnout), and QoL were assessed during hospitalization (i.e., the baseline). Furthermore, patients were asked to return to the hospital for follow-up examinations at 1, 6, and 12 months following discharge. QoL was evaluated at each follow-up.

The Ethics Committee of Kunming Medical University (Kunming, China) issued approval for this project and informed consent in written form was given by every participant.

### 2.2. Assessment of Work Stress, Anxiety, Depression, and Job Burnout

Two questionnaires, the Job Content Questionnaire (JCQ) and the ERI Questionnaire, were employed to determine work stress according to two theoretical models, respectively. The JCQ was developed by Karasek [5], and its Chinese version has been shown to have satisfactory reliability and validity [19]. The Chinese version of the ERI Questionnaire proposed by Siegrist [6] likewise shows good reliability and validity [20]. In this study, a short version of JCQ, which measures job demand and control by two items each, was adopted. This short version was demonstrated to be highly correlated with the complete version [21]. Similarly, in our study, a simplified ERI Questionnaire, which contained three effort-involving items plus six items for reward capture, was applied. The ERI Questionnaire in the shortened version is equivalent to the original version [22]. In this study, the Cronbach’s alpha coefficients for job demand, effort, job control, and reward were 0.77, 0.77, 0.75, and 0.79, respectively. The work stress levels were defined by virtue of a ratio of job demand to job control (with item numbers), or the effort/reward ratio (using item number for weighting) [23]. JS or ERI was dichotomized into two groups, low (ratio ≤ 1) and high (ratio > 1).

The Hospital Anxiety and Depression Scale (HADS) in Chinese was utilized to appraise depression and anxiety [24]: each subscale was rated between 0 and 21, and a higher score denoted more severe anxiety and depressive symptoms. The total HADS score is the sum of the anxiety score and depression score [25]. In this study, the HADS subscales for anxiety and depression produced Cronbach’s alpha coefficients of 0.71 and 0.73, respectively.

Based on the Chinese version of the Copenhagen Burnout Inventory (CBI), shown to be a trustworthy and effective tool, the job burnout subscale was employed to assess job burnout [26,27]. CBI scores range between 0 and 100, and higher scores indicate a greater degree of job burnout. The CBI job burnout subscale covering six items provided a Cronbach’s alpha coefficient of 0.72.

### 2.3. Evaluation of QoL

Two instruments were used to evaluate QoL: the Medical Outcome Study 8-Items Short Form Health Survey (SF-8) served as a generic instrument, while the Seattle Angina Questionnaire (SAQ) was taken as an instrument specific for CHD. In the eight-item SF-8 questionnaire, there are two component summary scales, correlated with physical and mental QoL [28]. Each scale has a total score of 0 to 100, and larger SF-8 scores signify better physical or mental QoL. Extensive application of the Chinese version of the SF-8 in has been recorded in patients with CHD in China [29]. In terms of the SF-8 physical and mental health dimensions in this study, the Cronbach’s alpha coefficients were 0.72 and 0.74, respectively. As a self-administered questionnaire with 19 items together with five scales, the SAQ is responsible for CHD measurement from the dimensions of angina frequency (AF), physical limitation (PL), disease perception (DP), angina stability (AS), and treatment satisfaction (TS) [30]. The higher the score on each scale (between 0 and 100), the more favorable the particular dimension. It has been demonstrated that the SAQ in Chinese possesses good validity and reliability in Chinese patients with CHD [31]. Cronbach’s alpha coefficients were obtained for PL (0.75), AS (0.71), AF (0.74), TS (0.76), and DP (0.70) dimensions of the SAQ in this study.

### 2.4. Statistical Analysis

Stata v.10 (Stata, College Station, TX, USA) was employed for statistical analysis, and alpha was set to 0.05 as the significance level. Mean ± standard deviation (SD) and number and percentage (%) were utilized to present the continuous variables and categorical variables, respectively. First, to compare the baseline characteristics between the groups with high and low work stress, the *t*-test and χ^2^ test were carried out to analyze the differences in continuous and categorical variables, respectively. Second, to analyze the changes of QoL scores over 1 year, repeated measures ANOVA was applied. Third, the longitudinal associations of work stress at the baseline with changes in QoL at varying time points (baseline, 1 month, 6 months, and 12 months) were determined by means of generalized estimating equations (GEE): the relationships among observations were processed through GEE under the conditions involving the same subjects [32,33]. Four steps were performed in the GEE regression modeling. Model I adjusted for demographic variables (age; sex; education level (primary school or lower, junior high school, high school, junior college/college, or above); number of family members; current smoking; household income per month; and heavy drinking) and clinical information (ACS type (UA, STEMI, and NSTEMI); Killip’s grade of cardiac function at admission; family history of cardiovascular diseases (CVD); and medical history (stroke, hypertension, dyslipidemia, and diabetes)) on patients at the baseline. Model II additionally adjusted for HADS, representing depression and anxiety, which are important CHD prognosis-associated risk factors, as previously recognized [34,35]. Model III was based on Model II with additional adjustment for job burnout, which is another notable factor contributing to changes in QoL among patients with ACS [36]. It should be noted that impacts of JS and ERI on QoL were selected for independent calculation, without adjustment for the other work stress model, across Models I to III. Finally, the association of the JS, ERI and QoL was estimated, with each work stress model adjusted for the others (Model IV). The purpose of such mutual adjustment was to examine how each model contributed independently to the longitudinal changes in QoL [23].

## 3. Results

### 3.1. Features of Participants at the Baseline

There were 150 eligible patients satisfying the inclusion criteria between March 2018 and December 2019, of whom 123 patients (82%) provided enrollment consent to this study. The baseline clinical data of one participant were incomplete because of medical electrode allergy; in addition, not all follow-up examinations were conducted for two patients. Consequently, this study recruited 120 patients (aged 27–62 years, with men and women numbering 101 and 19, respectively; median age: 50.5 years) for analysis. Cardiac events, including cardiac sudden death, heart failure, recurrent ACS events, and rehospitalization primarily induced by recurrent ACS-related symptoms, were not observed when the follow-up was terminated. Table 1 lists the participant features. Compared with the low JS group, the high JS group exhibited prominently raised scores for HADS and job burnout. In addition, the high effort–reward imbalance group differed from the low group in terms of the ACS type. Job burnout was higher in the high stress group. Other features, including age, sex, ACS type, Killip’s grade at admission, education level, family members, monthly income, smoking, drinking, medical history, family history of CVD, and medication, did not differ between the low and high JS groups, as well as the low and high ERI groups.

### 3.2. Change of QoL Scores over 1 Year after ACS

The QoL scores over 1 year are presented in Table 2, which reflects that the majority of QoL scores, including SF-8 physical health score, SF-8 mental health score, SAQ-PL, SAQ-TS, and SAQ-DP, gradually increased in the follow-up process lasting for 1 year, and they were lower in the high work stress group. However, the crude SAQ-AS and SAQ-AF scores did not show an obvious trend of gradual increase during the follow-up period.

### 3.3. Associations of Work Stress at the Baseline with Changes in QoL throughout 1 Year Post-ACS

As shown in Table 3, work stress at the baseline measured by JS was associated with slow recovery of the SF-8 physical health score after adjustment for demographic and clinical data, and baseline HADS score (both *p* < 0.05), but the physical health score in the group with high work stress displayed an insignificant difference in contrast to the low work stress group after adjusting for baseline job burnout (*p* = 0.076). JS-detected work stress was significantly related to slow recovery of the SF-8 mental health score after adjusting for demographic and clinical data, HADS score, and burnout score (all *p* < 0.05). Work stress based on JS was not associated with SAQ in terms of PL, AF, AS, DP, and TS (all *p* > 0.05). From the aspect of ERI at the baseline, work stress was related to slow improvement of the SF-8 physical health score, and to the mental health score in the case of adjustment for all confounders (all *p* < 0.01). The association of ERI-determined work stress with slow improvement of the SAQ-AS and SAQ-AF was significant after adjusting for all confounders (all *p* < 0.05). The ERI-determined work stress was related to slow improvement of the SAQ-PL after adjustment for demographic data and clinical data (*p* = 0.002), and adjustment for additional HADS score (*p* = 0.005), but after adjustment for job burnout, work stress was not associated with PL (*p* = 0.094). The association of ERI with changes in SAQ-TS and SAQ-DP, respectively, was found to be insignificant (all *p* > 0.05). Regarding independent contributions of JS and ERI to variations of QoL indicators, our data suggested that JS had no relation to any QoL parameter alteration (all *p* > 0.05). However, the correlations of ERI with slow recovery of the SF-8 physical health score, SAQ-AS, SF-8 mental health score, and SAQ-AF, respectively (all *p* < 0.05), still existed subsequent to mutual adjustment for the two models of work stress.

## 4. Discussion

Our study suggests that work stress predicts poor physical health and mental health recovery, and worse angina pectoris symptoms, and more frequent angina pectoris, one year after ACS in Chinese workers admitted to a hospital. The ERI seems to have stronger predictive power in the case of comparison between the two models of work stress. As far as we are aware, this study presents the longitudinal associations of work stress with changes in QoL among employed ACS patients from an Eastern country for the first time, and it is the first study comparing two work stress models in this regard.

According to existing reports, poor QoL is an independent risk factor for poor prognosis after ACS. The SF-36 [37] has become the most commonly applied generic questionnaire for QoL assessment. A 10-year prospective observational study showed that low SF-36 score predicted all-cause death in patients with myocardial infarction (MI) [38]. Low index scores for EuroQol-5 dimension (EQ-5D) [39], which is another generic tool to assess QoL, were found to predict adverse health outcomes after MI, such as all-cause death, recurrent MI, cardiovascular mortality, stroke, and unstable angina requiring urgent revascularization [40]. It has also been demonstrated that SAQ, as a measure of the disease-specific health status of patients with CHD, is independently associated with subsequent death, MI, and hospitalization [41,42,43]. Our study revealed the relationship between high baseline work stress and slow QoL recovery over time, including relatively low scores for physical and mental health, more angina pectoris symptoms, and more frequent angina pectoris. These observations imply that interventions targeting work stress mitigation may improve employees’ QoL after ACS and may thereby possibly reduce the risk of subsequent recurrence of cardiac events. In fact, cardiac rehabilitation and stress management programs have been demonstrated not only to decrease work stress but also to enhance QoL among workers [44], especially among those with ACS [45,46].

In terms of preventive action, it is of interest to compare the explanatory power between the two best-established work stress models. In general, both models have demonstrated complementary properties when explaining the etiology of chronic diseases such as depression [47] and CHD [9]. Furthermore, it has been suggested, based on findings from a Swedish study, to combine both models in order to improve risk estimation for CHD [48], and such an approach to CHD risk estimation was confirmed by a larger study including 11 cohorts from Europe [8]. In China, one study showed a slightly different pattern, that is, that ERI had a significant correlation with CHD in a sample of workers whereas JS did not [49]. Except for the German study by Kirchberger I et al. [15] and our present study from China, we are not aware of any other study examining links between work stress and subsequent QoL after the first episode of ACS. However, several studies conducted among apparently healthy workers without CHD have examined associations of the two work stress models with QoL. For instance, two cross-sectional studies from Brazil uncovered a closer relationship between the ERI and QoL mental and general health domains [50,51], and two longitudinal studies from Europe indicated a stronger capability of ERI to forecast QoL physical and mental health at follow-up [52,53]. In addition, our prior cross-sectional study suggested that ERI, as a work stress model, appeared to have more powerful explanatory ability for Chinese physicians’ QoL [23]. The current study, conducted in Chinese patients with ACS, indicated that ERI-associated high work stress was more powerful in predicting development of poor QoL after the onset of ACS, which was similar to the above cross-sectional and longitudinal studies [23,50,51,52,53]. Several explanations have been put forward to account for the differences between the two work stress models. The ERI model emphasizes that sufficient rewards are necessary to compensate the work-consumed efforts in a social exchange process. A considerably stressful situation occurs as long as such a norm of adequate exchange suffers violation because the sense of reciprocity and justification is impaired [6]. Enormous changes have appeared in current working life in China, which have led to high flexibility and mobility of jobs, the growth of short-term contracts and an increase of job insecurity [54,55]. The ERI, particularly the reward component, seems to become more important under these conditions, and is more sensitive to work stress among the Chinese working population than the JS model [23]. Moreover, one cultural difference characterized as individualism–collectivism deserves attention when interpreting work stress between western and eastern workers. This is particularly related to the role of job control: workers with individualist culture usually desire more personal autonomy, making a low level of job control as a salient source of work stress; whereas workers with collectivistic culture (from China, for instance) view themselves as interconnected within their workplace groups. Empirically, several studies suggested that explanatory power of job control on health was weaker among Asian workers [56].

After adjusting for baseline depression and anxiety, a general decrease was found in the absolute effect of GEE coefficients of work stress on changes in QoL (see Model II in Table 3). Depression and anxiety are common emotional responses to stressful life events (such as CHD). According to increasing numbers of studies, for patients with MI, depression and anxiety serve as independent risk factors leading to less ideal cardiac endpoints in the long run, and poorer QoL during the recovery from ACS [34,35,57,58]. Similarly, there was also a general decrease in the absolute value of GEE coefficients of work stress associated with changes in QoL after adjustment for baseline job burnout. This indicates that job burnout might mediate the effects of work stress on QoL. Job burnout refers to a condition of exhaustion resulting from increased time spent dealing with work-associated issues [59]. It is generally recognized from research on correlations involving work stress, disease, and job burnout that job burnout is a mediator for the relation of work stress to CHD and other health outcomes [60,61]. As reported from our previous findings, there is a correlation between baseline job burnout and low QoL amelioration rate after ACS [36]. Combined with the results of the present study, this further hints that job burnout plays a mediating role in the effects of CHD-related work stress plus health status.

In this study, solutions are required for several limitations. First, a fairly small sample size, especially the low proportion of female patients, probably makes the results less generalizable to the female subpopulation. However, in light of the enrollment standards set for our study (namely, employees experiencing their first ACS episode), the sample characteristics in this study seem reasonable. Second, work stress, depression, anxiety, and job burnout were assessed before discharge, and patients’ work status, work stress, depression, anxiety and burnout were not observed after discharge. However, ACS may change patients’ perceptions of work stress, anxiety, depression, and job burnout throughout the follow-up period. Consequently, the latent risk of exposure misclassification due to changes of employment status and work-related stress cannot be excluded, as recent research evidence indicated that more than 80% employees would return to work after coronary events [62], and work stress might be increased afterwards [63]. For future research, it is important to conduct dynamic observations on these participants to understand the relationship between the change in patients’ work stress and CVD prognosis. Third, data on sleep were not collected in our study, so we cannot rule out the impact of sleep factors on quality of life. Fourth, as observational research in a single center enrolling patients living in only one Chinese city, the generalizability of the results is restrained both inside and outside China.

## 5. Conclusions

In conclusion, our study indicates that work stress, particularly as measured by ERI, has a relationship with slow recovery of physical health and mental health, worse angina pectoris symptoms, and more frequent angina pectoris in patients with ACS across one year. The findings of our study add one piece of research evidence to current practice of cardiac rehabilitation programs that routinely apply measures of improving a healthy lifestyle. Yet, involvement of occupational health services is not widely considered into the cardiac rehabilitation [64]. In future, more attention should be paid to the role of work in terms of secondary prevention of CHD.

## Figures and Tables

**Table 1 ijerph-19-17018-t001:** Characteristics of subjects at the baseline (*n* = 120).

	Job Strain			Effort–Reward Imbalance		
	Low Group (*n* = 54)	High Group (*n* = 66)	*p*	Low Group (*n* = 52)	High Group (*n* = 68)	*p*
Age (years)	48.83 ± 8.42	50.05 ± 6.78	0.384	48.12 ± 7.99	50.56 ± 7.07	0.079
Male (*n*(%))	45 (83.33%)	56 (84.85%)	0.821	45 (86.54%)	56 (82.35%)	0.534
ACS type (*n*(%))						
UA	8 (14.81%)	9 (13.64%)	0.983	11 (21.15%)	6 (8.82%)	0.037
STEMI	25 (46.30%)	31 (46.97%)		18 (34.62%)	38 (55.88%)	
NSTEMI	21 (38.89%)	26 (39.39%)		23 (44.23%)	24 (35.29%)	
Killip’s grade of cardiac function at admission (n(%))						
Grade I	23 (42.59%)	30 (45.45%)	0.909	22 (42.31%)	33 (48.53%)	0.862
Grade II	21 (38.89%)	27 (40.91%)		22 (42.31%)	26 (38.24%)	
Grade III	8 (14.81%)	7 (10.61%)		7 (13.46%)	7 (10.29%)	
Grade IV	2 (3.71%)	2 (3.03%)		1 (1.92%)	2 (2.94%)	
Education level (*n*(%))						
Primary school or below	14 (25.93%)	20 (30.30%)	0.883	10 (19.23%)	24 (35.29%)	0.197
Junior high school	22 (40.74%)	25 (37.88%)		22 (42.31%)	25 (36.76%)	
High school	6 (11.11%)	9 (13.64%)		9 (17.31%)	6 (8.82%)	
Junior college/College or higher	12 (22.22%)	12 (18.18%)		11 (21.15%)	13 (19.12%)	
Number of family members (*n*)	3.91 ± 1.26	4.09 ± 1.36	0.448	3.90 ± 1.16	4.09 ± 1.42	0.448
Monthly family income (thousand Yuan)	7.144 ± 1.71	7.35 ± 2.46	0.588	7.40 ± 1.69	7.14 ± 2.45	0.520
Medical history (*n*(%))						
Hypertension	23 (42.59%)	27 (40.91%)	0.852	17 (32.69%)	33 (48.53%)	0.081
Diabetes	11 (20.37%)	17 (25.76%)	0.488	15 (28.85%)	13 (19.12%)	0.212
Dyslipidemia	23 (42.59%)	21 (31.82%)	0.223	23 (44.23%)	21 (30.88%)	0.133
Stroke	3 (5.56%)	3 (4.55%)	0.801	1 (1.92%)	5 (7.35%)	0.176
Family history of CVD	21 (38.89%)	18 (27.27%)	0.177	21 (40.38%)	18 (26.47%)	0.107
Current smoking	40 (74.07%)	41 (62.12%)	0.164	39 (75.00%)	42 (61.76%)	0.125
Heavy drinking	19 (35.19%)	20 (30.30%)	0.570	16 (30.77%)	23 (33.82%)	0.723
Medication [*n*(%)]						
Aspirin	54 (100.00%)	66 (100.00%)	-	52 (100.00%)	68 (100.00%)	-
P2Y12 receptor antagonists	54 (100.00%)	66 (100.00%)	-	52 (100.00%)	68 (100.00%)	-
Statin	54 (100.00%)	66 (100.00%)	-	52 (100.00%)	68 (100.00%)	-
Beta blockers	54 (100.00%)	66 (100.00%)	-	52 (100.00%)	68 (100.00%)	-
ACEI or ARB	54 (100.00%)	66 (100.00%)	-	52 (100.00%)	68 (100%)	-
HADS score	12.30 ± 4.49	14.59 ± 5.08	0.0107	12.85 ± 5.12	14.10 ± 4.76	0.168
Job burnout score	43.32 ± 17.01	54.40 ± 15.80	0.0003	39.50 ± 15.39	57.00 ± 14.51	<0.001

Continuous variables were presented by mean ± standard deviation (SD), and categorical variables were summarized as frequencies and percentages [n(%)]. Differences in continuous variables and categorical variables were examined via *t*-test and χ^2^ test, respectively. ACS: acute coronary syndrome; ACEI: angiotensin-converting enzyme inhibitor; UA: unstable angina; CVD: cardiovascular disease; ARB: angiotensin receptor antagonist; NSTEMI: non-ST-segment elevated myocardial infarction; STEMI: ST-segment elevated myocardial infarction; HADS: Hospital Anxiety and Depression Scale

**Table 2 ijerph-19-17018-t002:** The QoL scores over 1 year after ACS.

QoL Scores	Baseline	1 Month	6 Months	12 Months	*p* Value ^a^
SF-8 Physical health score	44.31 ± 6.84	46.20 ± 7.42	47.96 ± 6.41	48.82 ± 8.62	<0.001
Low JS	45.82 ± 6.92	46.86 ± 7.78	48.02 ± 6.58	49.21 ± 8.83	<0.001
High JS	43.18 ± 7.04	44.87 ± 7.54	47.23 ± 6.72	48.65 ± 9.09	<0.001
Low ERI	45.66 ± 6.96	46.92 ± 7.82	48.24 ± 7.08	49.33 ± 8.73	<0.001
High ERI	43.22 ± 7.02	44.30 ± 7.92	47.10 ± 6.92	48.23 ± 9.15	<0.001
SF-8 Mental health score	45.29 ± 7.62	46.45 ± 6.37	46.32 ± 5.87	48.25 ± 9.87	<0.001
Low JS	45.95 ± 7.96	47.13 ± 6.84	47.20 ± 6.27	48.77 ± 10.06	<0.001
High JS	44.10 ± 7.92	46.22 ± 7.22	46.11 ± 5.99	47.65 ± 10.02	<0.001
Low ERI	45.83 ± 7.75	47.21 ± 6.74	47.22 ± 5.96	48.54 ± 10.21	<0.001
High ERI	44.64 ± 7.97	46.17 ± 6.93	46.16 ± 6.07	47.23 ± 10.17	<0.001
SAQ PL	62.65 ± 10.32	66.05 ± 13.60	70.81 ± 8.51	72.96 ± 9.75	<0.001
Low JS	62.71 ± 10.53	66.28 ± 13.94	71.09 ± 8.92	73.12 ± 10.04	<0.001
High JS	62.52 ± 11.01	65.82 ± 14.19	70.57 ± 8.87	72.74 ± 10.32	<0.001
Low ERI	63.21 ± 10.98	66.89 ± 14.41	71.65 ± 8.75	73.97 ± 10.82	<0.001
High ERI	62.26 ± 10.68	65.32 ± 14.04	70.22 ± 8.98	72.36 ± 10.32	<0.001
SAQ AF	85.69 ± 14.53	77.95 ± 17.09	79.82 ± 11.74	80.61 ± 12.62	0.0029
Low JS	85.82 ± 14.68	78.05 ± 17.54	80.06 ± 11.97	80.94 ± 12.75	0.0339
High JS	85.32 ± 14.83	77.29 ± 17.85	79.47 ± 12.21	80.39 ± 12.88	0.029
Low ERI	86.13 ± 14.82	78.96 ± 17.22	80.22 ± 11.88	81.29 ± 12.90	0.1085
High ERI	85.23 ± 14.76	77.20 ± 17.92	79.19 ± 12.32	80.27 ± 12.93	0.0043
SAQ AS	92.82 ± 17.83	70.75 ± 25.14	74.45 ± 16.15	75.61 ± 18.19	0.4643
Low JS	92.90 ± 18.33	70.80 ± 26.54	74.66 ± 16.35	75.32 ± 19.42	0.8669
High JS	92.73 ± 18.12	70.22 ± 27.33	74.23 ± 16.58	75.70 ± 19.20	0.4696
Low ERI	93.45 ± 18.05	71.43 ± 25.28	75.08 ± 16.78	75.97 ± 18.79	0.175
High ERI	92.03 ± 18.13	70.16 ± 26.09	74.19 ± 16.35	74.88 ± 19.08	0.8399
SAQ TS	68.82 ± 12.63	69.12 ± 14.43	74.65 ± 12.10	74.39 ± 13.33	<0.001
Low JS	69.05 ± 13.21	69.51 ± 14.59	74.78 ± 12.33	74.55 ± 13.49	<0.001
High JS	68.32 ± 13.09	68.89 ± 14.69	74.21 ± 12.29	74.17 ± 13.52	0.0031
Low ERI	69.10 ± 12.87	69.56 ± 14.82	74.81 ± 12.54	74.76 ± 13.55	0.0049
High ERI	68.48 ± 13.00	69.05 ± 14.78	74.32 ± 12.33	74.20 ± 13.48	0.002
SAQ DP	45.15 ± 17.75	63.75 ± 18.98	78.74 ± 20.61	80.65 ± 19.72	<0.001
Low JS	45.38 ± 18.06	64.09 ± 19.22	78.89 ± 20.87	81.13 ± 20.20	0.0068
High JS	44.88 ± 18.12	63.33 ± 19.10	78.32 ± 21.17	80.38 ± 20.22	<0.001
Low ERI	45.69 ± 18.19	64.23 ± 19.27	79.08 ± 21.10	81.20 ± 19.86	0.005
High ERI	44.68 ± 17.89	63.52 ± 19.17	78.56 ± 21.31	80.42 ± 20.12	<0.001

^a^ Repeated measures ANOVA. Time trends of QoL scores were examined by repeated measures ANOVA. JS: job strain; ERI: effort–reward imbalance; DP: disease perception; PL: physical limitation; SF-8: Medical Outcome Study 8-Items Short Form Health Survey; TS: treatment satisfaction; AF; angina frequency; SAQ: Seattle Angina Questionnaire; AS: angina stability. All QoL scores were expressed as mean ± standard deviation (SD).

**Table 3 ijerph-19-17018-t003:** The coefficients and 95% CIs of repeated measures of QoL parameters during the one-year follow-up by work stress at the baseline according to job strain and effort–reward imbalance.

Work Stress		Model I		Model II		Model III		Model IV	
		Coefficient (95% CIs)	*p* Value	Coefficient (95% CIs)	*p* Value	Coefficient (95% CIs)	*p* Value	Coefficient (95% CIs)	*p* Value
		SF-8—physical health score							
JS	Low	0.00		0.00		0.00		0.00	
	High	−3.56 (−5.69, −1.44)	0.001	−2.81 (−4.99, −0.63)	0.011	−1.88 (−3.96, 0.20)	0.076	−1.29 (−3.26, 0.68)	0.20
ERI	Low	0.00		0.00		0.00		0.00	
	High	−6.32 (−8.37, −4.27)	<0.001	−5.87 (−7.90, −3.83)	<0.001	−4.70 (−6.90, −2.51)	<0.001	−4.49 (−6.69, −2.28)	<0.0001
		SF-8—mental health score							
JS	Low	0.00		0.00		0.00		0.00	
	High	−3.51 (−5.42, −1.59)	<0.001	−2.49 (−4.38, −0.61)	0.009	−1.99 (−3.87, −0.11)	0.038	−1.59 (−3.43, 0.25)	0.091
ERI	Low	0.00		0.00		0.00		0.00	
	High	−4.56 (−6.54, −2.57)	<0.001	−3.87 (−5.74, −2.00)	<0.001	−3.30 (−5.36, −1.23)	0.002	−3.03 (−5.09, −0.97)	0.004
		SAQ—PL							
JS	Low	0.00		0.00		0.00		0.00	
	High	−3.70 (−7.56, 0.16)	0.060	−2.93 (−6.92, 1.05)	0.149	−1.64 (−5.61, 2.32)	0.416	−1.16 (−5.15, 2.83)	0.567
ERI	Low	0.00		0.00		0.00		0.00	
	High	−6.32 (−10.37, −2.27)	0.002	−5.82 (−9.89, −1.75)	0.005	−3.80 (−8.23, 0.64)	0.094	−3.60 (−8.07, 0.88)	0.115
		SAQ—AS							
JS	Low	0.00		0.00		0.00		0.00	
	High	−4.07 (−10.68, 2.53)	0.227	−1.98 (−8.77, 4.80)	0.567	−0.83 (−7.83, 6.17)	0.816	0.44 (−6.52, 7.41)	0.901
ERI	Low	0.00		0.00		0.00		0.00	
	High	−11.46 (−18.41, −4.52)	0.001	−10.22 (−17.18, −3.27)	0.004	−9.84 (−17.61, −2.08)	0.013	−9.92 (−17.76, −2.07)	0.013
		SAQ—AF							
JS	Low	0.00		0.00		0.00		0.00	
	High	−3.81 (−8.29, 0.67)	0.096	−2.09 (−6.59, 2.40)	0.361	−1.43 (−6.03, 3.18)	0.544	−0.69 (−5.24, 3.87)	0.768
ERI	Low	0.00		0.00		0.00		0.00	
	High	−7.03 (−11.66, −2.39)	0.003	−5.97 (−10.50, −1.43)	0.010	−5.60 (−10.65, −0.54)	0.030	−5.48 (−10.60, −0.36)	0.036
		SAQ—TS							
JS	Low	0.00		0.00		0.00		0.00	
	High	−1.31 (−6.02, 3.39)	0.585	−0.11 (−4.97, 4.74)	0.964	1.50 (−3.33, 6.32)	0.544	1.47 (−3.41, 6.34)	0.555
ERI	Low	0.00		0.00		0.00		0.00	
	High	−3.32 (−8.34, 1.69)	0.194	−2.55 (−7.61, 2.52)	0.324	0.46 (−5.02, 5.94)	0.870	0.23 (−5.29, 5.75)	0.934
		SAQ—DP							
JS	Low	0.00		0.00		0.00		0.00	
	High	4.27 (−2.66, 11.19)	0.227	4.70 (−2.33, 11.73)	0.190	5.70 (−1.87, 13.28)	0.140	7.35 (−0.14, 14.84)	0.055
ERI	Low	0.00		0.00		0.00		0.00	
	High	−2.25 (−10.21, 5.72)	0.581	−2.44 (−10.54, 5.66)	0.555	−2.57 (−10.05, 4.92)	0.502	2.71 (−4.85, 10.26)	0.483

JS: job strain; ERI: effort–reward imbalance; CIs: confidence intervals; ACS: acute coronary syndrome; CVD: cardiovascular disease; STEMI: ST-segment elevated myocardial infarction; UA: unstable angina; NSTEMI: non-ST-segment elevated myocardial infarction; HADS: Hospital Anxiety and Depression Scale; AF: angina frequency; SF-8: Medical Outcome Study 8-Items Short Form Health Survey; PL: physical limitation; AS: angina stability; SAQ: Seattle Angina Questionnaire; TS: treatment satisfaction; DP: disease perception. Generalized estimating equations. Model I: adjusted for age, sex, ACS type (UA, STEMI, NSTEMI), Killip’s grade of cardiac function at admission, education level (primary school or below, junior high school, high school, junior college/College or higher), number of family members, monthly family income, medical history (hypertension, diabetes, dyslipidemia, stroke), family history of CVD, current smoking, heavy drinking. Model II: Model I + additionally adjusted for HADS score. Model III: Model II + additionally adjusted for job burnout score. Model IV: Model III + job strain and effort–reward imbalance were mutually adjusted for each other.

## Data Availability

All data generated or analyzed during the study appear in the submitted article.

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
