# Peer review of "Longitudinal Associations of Work Stress with Changes in Quality of Life among Patients after Acute Coronary Syndrome: A Hospital-Based Study"

_ijerph, 2022, doi:10.3390/ijerph192417018_

Round 1
Reviewer 1 Report
Although the present study is generally well done and well-written, it needs to be modified or amended before accepting the manuscript.
1. In the Introduction, you have mentioned about the potential cultural difference between Germany and China. Can you elaborate more on this topic in the discussion?
2. Is there any way to control the severity of ACS in the statistical model, although you have controlled for ACS types.
3. Can you additionally control for sleep factors and physical activity? If you cannot, please mention it on the limitation section.
4. Were these patients working at the same workplace or engaged in the same occupation after ACS episode? Or have they quit their job or changed their job?
5. There may be sex differences in the results because job stress is different between men and women in the eastern culture context. Were the results consistent when excluding women? Can you consider sensitivity analysis?
6. I think when you try to compare the magnitude of the effects of ERI and JS on outcome variables, you may need to standardize the ERI and JS values.
Author Response
Dear Reviewer:
Thank you for your helpful comments and suggestions. Based on your comments, our responses are addressed below:
Point 1: In the Introduction, you have mentioned about the potential cultural difference between Germany and China. Can you elaborate more on this topic in the discussion?
Response 1: In the 3rd paragraph of the “Discussion” section, we expanded the cultural difference and a new reference was added.
Several explanations have been put forward to account for the differences between the two work stress models. The ERI model emphasizes that sufficient rewards are necessary to compensate the work-consumed efforts in a social exchange process. A considerably stressful situation occurs as long as such a norm of adequate exchange suffers violation because the sense of reciprocity and justification is impaired [6]. Enormous changes have appeared in current working life in China, which have led to high flexibility and mobility of jobs, the growth of short-term contracts and an increase of job insecurity [54, 55]. The ERI, particularly the reward component, seems to become more important under these conditions, and is more sensitive to work stress among the Chinese working population than the JS model [23]. Moreover, one cultural difference characterized as individualism–collectivism deserves attention when interpreting work stress between western and eastern workers. This is particularly related to the role of job control: workers with individualist culture usually desire more personal autonomy, making a low level of job control as a salient source of work stress; whereas workers with collectivistic culture (from China, for instance) view themselves as interconnected within their workplace groups. Empirically, several studies suggested that explanatory power of job control on health was weaker among Asian workers [56].
- Mazzola JJ, Schonfeld IS, Spector PE: What qualitative research has taught us about occupational stress. Stress Health 2011, 27(2):93-110.
Point 2: Is there any way to control the severity of ACS in the statistical model, although you have controlled for ACS types.
Response 2: We did collect data on cardiac function of ACS patients at admission, i.e. Killip’s grade of cardiac function, which reflected the severity of ACS. In the regression analyses, Killip’s grade has been included in the model 1, although we did not particularly emphasize it.
Point 3: Can you additionally control for sleep factors and physical activity? If you cannot, please mention it on the limitation section.
Response 3: Thank you for pointing this out. We did not collect the data on sleep, and we have mentioned it in the limitation section in the revised manuscript.
Third, data on sleep were not collected in our study, so we cannot rule out the impact of sleep factors on quality of life.
As for physical activity, SAQ-PL and SF-8 physical health score indirectly reflects the patients' capacity of physical activity, as two outcomes of this study.
Point 4: Were these patients working at the same workplace or engaged in the same occupation after ACS episode? Or have they quit their job or changed their job?
Response 4: We did not have information on patients’ work status and work stress after discharge, and we added it in the limitation section as this:
Second, work stress, depression, anxiety, and job burnout were assessed before discharge, and patients’ work status, work stress, depression, anxiety and burnout were not observed after discharge. But ACS may change patients' perceptions of work stress, anxiety, depression, and job burnout throughout the follow-up period. Consequently, the latent risk of exposure misclassification due to changes of employment status and work-related stress cannot be excluded, as recent research evidence indicated that more than 80% employees would return to work after coronary events [62], and work stress might be increased afterwards [63]. For future research, it is important to conduct dynamic observations on these participants to understand the relationship between the change in patients' work stress and CVD prognosis.
Point 5: There may be sex differences in the results because job stress is different between men and women in the eastern culture context. Were the results consistent when excluding women? Can you consider sensitivity analysis?
Response 5: We additionally conducted sensitivity analysis by excluding women participants (see attached supplementary Table 1). There were fewer female participants in this study, and the results were consistent with current findings excluding women participants. The subjects of this study was ACS patients of working age. At this age, most women are still in the reproductive period. Because of the cardioprotective effect of estrogen, the proportion of women with coronary heart disease is very low. Therefore, we said in the limitation section, "probably makes the results less generalizable to the female subpopulation. However, in light of the enrollment standards set for our study (namely, employees experiencing their first ACS episode), the sample characteristics in this study seem reasonable. ”
Point 6: I think when you try to compare the magnitude of the effects of ERI and JS on outcome variables, you may need to standardize the ERI and JS values.
Response 6: We agree that standardization of ERI and JS values would be needed when using continuous scores of JS or ERI in the regression modeling. We need to note that, in this study, two binary/dichotomized variables of high vs. low JS and ERI were applied. Therefore, we did not standardize the ERI and JS values.

Reviewer 2 Report
[A brief summary]
The manuscript concerns a longitudinal study in 120 Chinese employees who treated their first episode of acute coronary syndrome (ACS) in a large university hospital in China, focusing on the effect of work stress on longitudinal quality of life (QoL) variations after discharge.
In short, work stress was associated with slow recovery of QoL for patients with ACS. More specifically, both job strain (JS) and effort reward imbalance (ERI) models predicted slow recovery of QoL measured by 8-Items Short Form (SF-8), and ERI model independently had the larger predictive power than JS model. In addition, ERI-determined work stress affected slow recovery of QoL in anginal stability (AS) and anginal frequency (AF) scales of Seattle Angina Questionnaire (SAQ).
Thus, this article highlights work stress for patients with ACS as a predictor of QoL variations. The manuscript is well written, include relevant figures, is short and to the point.
[Strengths]
- Using hospital-based data, the possibility of misclassification of diagnosis of ACS is low.
- A longitudinal study design prevented reverse causality.
- More than one instruments were used in the measurement of exposure and end point.
- Potential confounding variables were properly controlled.
[Weakness]
Results
3.1 Features of participants at baseline
This section does not include all the results shown in Table 1 (e.g., education level, family members, etc.). Please summarize all the results in Table 1, as briefly as possible.
3.2 Change of QoL scores over 1 year after ACS
This part does not seem to adequately summarize the results of Table 2. For example, SAQ AF and SAQ AS scores were not gradually increase during one-year follow-up. According to reference 30, AS and AF scale scores were estimated based on question 2 and question 3-4 in SAQ, respectively. I understand that the higher the score in each SAQ scale means more favorable function in ACS patients. In table 2, the gradually decrease in SAQ AF score in high ERI group cannot indicate recovery of QoL.
Discussion
As the authors point out clearly, this study did not consider the effect of ACS episode on work stress and potential confounding and mediating factors after discharge. After discharge, participants’ employment status or working environment such as working hours, work schedules, interpersonal relationship and income may have changed voluntarily or involuntarily according to his/her health status, which may lead to misclassification of work stress groups based on the JS and ERI models. Please consider further analysis improving the weakness, or describe what efforts have been made to compensate for the limitation, or describe what kind of study design is needed for further studies to compensate for this methodological limitation.
Conclusions
There is a logical gap in suggesting that the results of this study emphasize an effect of work-related factors on CHD prognosis; logical gap in post-ACS patients’ QoL versus CHD prognosis. In addition, both ERI- and JS- determined work stress were not related to SAQ-TS and SAQ-DP.
[Tables]
All tables should be self-explanatory. Please indicate whether the figures in table 1 and table 2 are the mean and standard (mean±SD) or number and % (n(%)). Please define all acronyms used in the table (e.g., ACS in table 2) at the table legend. In table 3, it should be indicated which variables were adjusted for each GEE regression models.
In table 1, please consider more clearly expressing the unit of age variable as Age (yrs) or Age (years) rather than Age (y).
Author Response
Dear Reviewer:
Thank you for your instructive comments and suggestions which have helped us to further strengthen our manuscript. Our responses are addressed below:
Point 1: in “3.1 Features of participants at baseline” of “Results” section, you pointed that: This section does not include all the results shown in Table 1 (e.g., education level, family members, etc.). Please summarize all the results in Table 1, as briefly as possible.
Response 1: We added one sentence at the end of "3.1 Features of participants at baseline", and they are displayed in red in the revised manuscript:
3.1 Features of participants at baseline
Other features, including age, sex, ACS type, Killip’s grade at admission education level, family members, monthly income, smoking, drinking, medical history, family history of CVD, and medication, did not differ between the low and high JS groups, as well as the low and high ERI groups.
Point 2:
3.2 Change of QoL scores over 1 year after ACS
This part does not seem to adequately summarize the results of Table 2. For example, SAQ AF and SAQ AS scores were not gradually increase during one-year follow-up. According to reference 30, AS and AF scale scores were estimated based on question 2 and question 3-4 in SAQ, respectively. I understand that the higher the score in each SAQ scale means more favorable function in ACS patients. In table 2, the gradually decrease in SAQ AF score in high ERI group cannot indicate recovery of QoL.
Response 2: Thank you for pointing this out. We have revised the text description of Table 2 as below:
3.2 Change of QoL scores over 1 year after ACS
The QoL scores over 1 year are presented in Table 2, which reflects that the majority of QoL scores, including SF-8 physical health score, SF-8 mental health score, SAQ-PL, SAQ-TS, and SAQ-DP, gradually increased in the follow-up process lasting for 1 year, and they were lower in the high work stress group. However, the crude SAQ-AS and SAQ-AF scores did not show an obvious trend of gradual increase during the follow-up period.
Point 3:
Discussion
As the authors point out clearly, this study did not consider the effect of ACS episode on work stress and potential confounding and mediating factors after discharge. After discharge, participants’ employment status or working environment such as working hours, work schedules, interpersonal relationship and income may have changed voluntarily or involuntarily according to his/her health status, which may lead to misclassification of work stress groups based on the JS and ERI models. Please consider further analysis improving the weakness, or describe what efforts have been made to compensate for the limitation, or describe what kind of study design is needed for further studies to compensate for this methodological limitation.
Response 3: We are grateful for your suggestion, we have made some supplementary explanations in the “limitation” section and added two references:
Second, work stress, depression, anxiety, and job burnout were assessed before discharge, and patients’ work status, work stress, depression, anxiety and burnout were not observed after discharge. But ACS may change patients' perceptions of work stress, anxiety, depression, and job burnout throughout the follow-up period. Consequently, the latent risk of exposure misclassification due to changes of employment status and work-related stress cannot be excluded, as recent research evidence indicated that more than 80% employees would return to work after coronary events [62], and work stress might be increased afterwards [63]. For future research, it is important to conduct dynamic observations on these participants to understand the relationship between the change in patients' work stress and CVD prognosis.
Point 4:
Conclusions
There is a logical gap in suggesting that the results of this study emphasize an effect of work-related factors on CHD prognosis; logical gap in post-ACS patients’ QoL versus CHD prognosis. In addition, both ERI- and JS- determined work stress were not related to SAQ-TS and SAQ-DP.
Response 4: Thanks for your suggestion. In the revised manuscript, we have modified conclusions as below:
In conclusion, our study indicates that work stress, particularly as measured by ERI, has a relationship with slow recovery of physical health and mental health, worse angina pectoris symptoms, and more frequent angina pectoris in patients with ACS across one year. The findings of our study add one piece of research evidence to current practice of cardiac rehabilitation programs that routinely apply measures of improving a healthy lifestyle. Yet, involvement of occupational health services is not widely considered into the cardiac rehabilitation [64]. In future, more attention should be paid to the role of work in terms of secondary prevention of CHD.
Point 5:
[Tables]
All tables should be self-explanatory. Please indicate whether the figures in table 1 and table 2 are the mean and standard (mean±SD) or number and % (n(%)). Please define all acronyms used in the table (e.g., ACS in table 2) at the table legend. In table 3, it should be indicated which variables were adjusted for each GEE regression models.
In table 1, please consider more clearly expressing the unit of age variable as Age (yrs) or Age (years) rather than Age (y).
Response 5: We have revised Table 1, Table 2 and Table 3 according to your comments.
Reviewer 3 Report
The paper aims to investigate whether and how work stress can be correlated with quality of life in patients suffering from acute coronary syndrome.
The authors have studied Chinese employees over 12 months, evaluating their work stress and quality of life levels in different moments, through the use of specific questionnaires.
The analysis of the collected data suggests that high levels of work stress can affect quality of life, as results show lower scores for physical and mental health.
In general, the contribution has a clear structure, composed of different sections that facilitate reading. I would suggest a thorough proof-reading, though.
Title and abstract serve their purpose.
In the introduction, the problem is explained and the approach to the study is set. It should be better if appropriate research questions were clearly defined.
The methodological approach is solid and based on reliable tools.
Results are clearly shown and tables are helpful.
References are accurate and take into account relevant and recent studies.
Conclusion is too short and needs to be expanded.
Author Response
Dear Reviewer,
Thank you for your helpful suggestions. We have revised our manuscript according to your suggestion as below:
- In the introduction, it should be better if appropriate research questions were clearly defined.
Response: In the revised version, we rewrote “the aim of our study” to make the research questions clearer:
In the single prior study [15], only ERI model was used to access work stress, and the participants of the study were from a Western country (i.e., Germany). The aim of our study was to examine prospective links of work stress with changes in QoL among patients after ACS, by expanding insights from the single prior study [15] in two ways. First, in addition to the ERI we examined the JS model and compared the two models of work stress from the angle of their explanatory contributions. Those analyses can inform the development of interventions that target specific psychosocial work characteristics to improve QoL in patients ACS. Second, novel research from an Eastern country (i.e., China) is provided by our study, highlighting obvious cultural differences in work stress [16].
- Conclusion is too short and needs to be expanded.
Response: We have expanded the section of “conclusion” as below:
In conclusion, our study indicates that work stress, particularly as measured by ERI, has a relationship with slow recovery of physical health and mental health, worse angina pectoris symptoms, and more frequent angina pectoris in patients with ACS across one year. The findings of our study add one piece of research evidence to current practice of cardiac rehabilitation programs that routinely apply measures of improving a healthy lifestyle. Yet, involvement of occupational health services is not widely considered into the cardiac rehabilitation [64]. In future, more attention should be paid to the role of work in terms of secondary prevention of CHD.
Round 2
Reviewer 1 Report
I think my major concern well appropriately answered and now it is in a good shape.
Reviewer 2 Report
You have responded to all of my comments and revised the manuscript accordingly. I have no further comments.